# Health-related quality of life of adults with cutaneous leishmaniasis at ALERT Hospital, Addis Ababa, Ethiopia

**Shimelis Doni**[1☯]*, **Kidist Yeneneh**[2☯], **Yohannes Hailemichael**[3], **Mikyas Gebremichael**[3], **Sophie Skarbek**[4], **Samuel Ayele**[3], **Abay Woday Tadesse**[3], **Saba Lambert**[1,5], **Stephen L. Walker**[5], **Endalamaw Gadisa**[3]

**1** ALERT Hospital, Addis Ababa, Ethiopia, **2** Addis Ababa University, Addis Ababa, Ethiopia, **3** Armauer Hansen Research Institute, NTD and Malaria Research Directorate, Addis Ababa, Ethiopia, **4** Leeds Teaching Hospitals NHS Foundation Trust, Leeds, United Kingdom, **5** Faculty of Infectious and Tropical Diseases, London School of Hygiene and Tropical Medicine, London, United Kingdom

☯ These authors contributed equally to this work.

\* shimelis321@gmail.com

**Data Availability Statement:** We have presented all relevant data as a figure and/or table within the manuscript. The remaining data have personal

## Abstract

### Background

Cutaneous leishmaniasis (CL) is a growing public health threat in Ethiopia. *Leishmania aethiopica* is the predominant causative organism. Affected individuals develop chronic skin lesions on exposed parts of the body, mostly on the face, which are disfiguring and cause scarring. The effects of CL on the health-related quality of life (HRQoL) of affected individuals has not been assessed in Ethiopia.

### Objective

To assess HRQoL in adults with active CL at ALERT Hospital, Addis Ababa, Ethiopia.

### Methods

A cross-sectional study was done using the Amharic version of the Dermatology Life Quality Index (DLQI). Trained health staff administered the DLQI.

### Results

Three hundred and two adults with active CL participated and all of them exhibited a reduced HRQoL. The median DLQI score was 10 (IQR 8). Almost half of the participants reported very poor HRQoL, 36.4% and 11.3% fell within the very large and extremely large effect categories respectively. DLQI scores were higher (median 18) in patients diagnosed with diffuse cutaneous leishmaniasis (DCL) compared to those with localized cutaneous leishmaniasis (LCL). The DLQI domain of 'work and school' was the most affected, scoring 73.3% and 66.6% of total possible score for female and male respectively, followed by that of 'symptom and feeling' (at 50.0% and 56.6% for female and male respectively). Men were more affected than women in the domains of 'leisure' (P = 0.002) and 'personal

participant data for which we have no consent or ethical clearance to share data in its entirety. Request for data used in this manuscript in Stata or Any other data software format could be made to the AHRI-ALERT ethics committee "ahri. alerterc@ahri.gov.et.

**Funding:** This work was funded by the Armauer Hansen Research Institute (Norad and Sida) Core funding. The funder had no role in study design, data collection and analysis, decision to publish, or preparation of the manuscript.

**Competing interests:** The authors have declared that no competing interests exist.

relationships' (P = 0.001). In the multivariate ordinal logistic regression site of lesion, clinical phenotype and age of participant remained associated with significantly poor HRQoL.

## Conclusion

The HRQoL impairment associated with CL is significant. Thus, patient-reported outcome measure should be used to assess the efficacy of treatments along with clinical outcome measures.

## Author summary

Cutaneous leishmaniasis (CL) is an important public health problem in Ethiopia with an estimated incidence up to 50,000 cases per year. CL is predominately due to *Leishmania aethiopica*. The transmission is by sandflies, with hyraxes being reservoir hosts. Lesions are chronic on the exposed part of the body, commonly on the face. Three main clinical phenotypes are recognized; localized (LCL), muco-cutaneous (MCL), and diffuse (DCL) cutaneous leishmaniasis. The permanent damage and altered anatomy of the skin, nose, eyelids, ears and lips due to scarring is often associated with stigma.

This study aimed to assess the health-related quality of life (HRQoL) using the Amharic version of the Dermatology Life Quality Index (DLQI) in adults diagnosed with active CL. Trained interviewers administered the DLQI to participants prior to treatment. Our results show that the impact on HRQoL associated with CL is large. There was no significant difference between men and women, and urban and rural dwellers. Participants with DCL, the more extensive form of CL, had lower HRQoL compared to those with other forms. Those with lesions on their head and neck regions and younger than 50 years (20 to 49 years age group) had significantly lower HRQoL.

The observed HRQoL associated with CL calls for improved support for affected individuals as part of CL care and treatment to improve outcomes. Importantly, patient-reported outcome measures including the DLQI should be used to assess the efficacy of treatments.

## Introduction

The leishmaniases are neglected tropical diseases caused by an intracellular protozoan of the genus *Leishmania* transmitted by the bite of infected female phlebotomine sandflies. Cutaneous leishmaniasis (CL) is the most common form of leishmaniasis with an estimated million new cases per year globally [1]. CL, caused by *Leishmania aethiopica (L. aethiopica)*, is a major public health problem in Ethiopia with over 29 million people estimated to be at risk and up to 50000 new cases per year [2]. Affected individuals develop chronic, granulomatous cutaneous lesions on exposed parts of the body, predominantly the face. Lesions heal with scarring [3–5]. Children and young adults are the most affected [5–7]. The clinical phenotypes of CL due to *L. aethiopica* are localized (LCL), mucocutaneous (MCL) and diffuse (DCL) [8,9]. MCL and DCL present significant therapeutic challenges.

CL has a severe psychosocial and economic impact, which contributes to perpetuating poverty. CL is associated with reduced health-related quality of life (HRQoL) of affected individuals due to the appearance of active skin lesions or the permanent scarring on exposed body

sites [10,11]. HRQoL encompasses physical activity, psychological well-being, degree of independence and social relationships, all of which are affected by skin disease [12]. Studies in other settings have shown affected individuals with CL have reduced quality of life. Boukthir et al., in a Tunisian study reported the impact of CL that ranged from scar related stigma to suicidal thoughts from fear of being disabled and stress caused by close relatives [10]. Khatami et al.documented feelings of sadness, guilt, anxiety, stigma and exclusion through indepth interviews of 12 Iranian individuals with CL [11]. CL is associated with a negative impact on personal relationships and self-esteem resulting in anxiety and depression. Bennis et al., highlighted that CL could adversely affect life choices such as marriageability in endemic communities [13].

The Dermatology Life Quality Index (DLQI) is used to assess the HRQoL of individuals with CL in Iran [14], Brazil [15] and the United Kingdom [16]. The proportion of individuals with a moderate or greater impact of CL on their HRQoL ranged from 42.7 to 70%. The severity, clinical phenotype, and sociodemographic characteristics of individuals appears to influence HRQoL associated with CL [17]. There is paucity of data on HRQoL of individuals with CL in Ethiopia. We aimed to assess the HRQoL associated with active CL

## Methods

### Ethical consideration

This study was reviewed and approved by AHRI/ALERT Ethics Review Committee (approval number: PO/43/18). The participant information sheet and consent were in Amharic. Written informed consent was obtained from all participants.

### Study setting

This work was done at the dermatology department of ALERT Hospital, Addis Ababa, Ethiopia.

### Study population and data collection

A cross-sectional study was performed between December 2018 and December 2021. Individuals aged 18 years or older diagnosed with active CL (confirmed by microscopy and/or culture) who gave written informed consent were enrolled. Inclusion was not consecutive as staff were not always available to recruit and not all affected individuals spoke Amharic.

The treating health care professionals classified participants as having LCL, MCL or DCL. Demographic and clinical data including location of lesions were recorded on a data collection form. The sites of skin lesions were categorised into regions as being on the head and neck or the torso and/or limbs or both regions. The head and neck skin lesions were further categorised into those affecting the face (not the lips or nose), lips or nose.

The validated Amharic version of the DLQI [18] was completed for each participant prior to treatment by a trained interviewer.

The DLQI is a 10-item questionnaire addressing six aspects of life (domains); symptoms and feelings (Questions 1 and 2), daily activities (Questions 3 and 4), leisure (Questions 5 and 6), work and school (Question7), personal relationships (Questions 8 and 9), and treatment (Question 10). The scores to each of the 10 items add up giving a total score ranging from 0 to 30. A higher DLQI score indicates greater impairment of HRQoL.

Individuals rate the impact of their dermatological condition in the past week as "not at all, scored 0", "a little, scored 1", "a lot, scored 2", "very much, scored 3", "not relevant, scored 0" or Question 7, 'prevented work or studying' scored 3". The HRQoL impact is interpreted

using the total DLQI score as 0–1 no effect at all, 2–5 small effect, 6–10 moderate effect, 11–20 very large effect and 21–30 extremely large effect. The scores for each domain are expressed as a percentage of the total possible score for the domain.

## Data management and analysis

Data were checked for completeness and double entered into *EpiData* (version 3.1, *EpiData* Association, Odense, Denmark), exported and analyzed using *Stata* (version 17.0, *Stata* Corporation, College Station, TX, USA). For data analysis participants were grouped according to sex (male or female), age (in years), residence (urban or rural identified according to Kebele of residence), body region affected and CL clinical phenotype. Participants with missing data were to be excluded but no data were missing.

A comparison of scores between two groups was made using the Mann–Whitney U inspected rank test, and for three or more groups the Kruskal–Wallis H test. Adjusted multivariable ordinal logistic regression analysis was performed to identify the independent predictors of effect of CL on HRQoL. We used a model based on clinical phenotype, location of lesions, sex, age and residence P-values less than 0.05 were considered statistically significant.

## Results

### Characteristics of the participants

Three hundred and two individuals diagnosed with active CL were recruited. The median age of participants was 29 years (IQR 21; 45), 56.0% male (169/302) and 66.6% urban dwellers (201/302). The body region most affected was the head and neck in 270 (89.4%), of which involve the nose in 151 (50%), other sites on the face in 122 (40.4%), and the lips in 35 (11.6%). The proportion of clinical phenotypes were 62.6% LCL (189/302), 34.4% MCL (104/302), and 3.0% DCL (9/302) (Table 1).

Males accounted for 55.5% (104/189), 55.77% (58/104) and 77.78% (7/9) of the LCL, MCL and DCL cases respectively (Table 2).

### DLQI scores of participants

The overall median DLQI score for the study participants was 10 (IQR 8). The range of DLQI scores was 2 to 29. Median DLQI scores were higher in participants diagnosed with DCL (median 18) compared to participants with MCL (median 11) and LCL (median 9) (Table 1 and Fig 1). Similarly, participants in the 30–39 year age group had higher DLQI scores (median 14; IQR 9).

### Size of HRQoL effect associated with active cutaneous leishmaniasis

The size of the HRQoL effect of active CL as measured by DLQI score ranged from small effect to extremely large effect (Table 3). Almost half of the participants reported markedly reduced HRQoL, 36.4% and 11.3% fell within the very large and extremely large effect categories respectively. In addition, 39.4% of the participants experienced a moderate effect. All the nine individuals with DCL had a DLQI score of 10 or more (Fig 1).

### DLQI domains

The percentage of domain scores were calculated from the total possible score. The percentage and the medians with interquartile ranges for each of the six domains by sex is shown in Table 4. The 'work and school' domain had the highest (73.3% and 66.6% for females and males respectively) total possible percentage scores whereas the 'personal relationships'

**Table 1. Sociodemographic, disease characteristics and Dermatology Life Quality Index (DLQI) scores of participants.**

| Variables | Number (%) | DLQI Score | |
|---|---|---|---|
| | | Median | IQR |
| **Age (years)** | | | |
| 18–19 | 46 (15.2) | 9.5 | 10.0 |
| 20–29 | 106 (35.1) | 11 | 8.0 |
| 30–39 | 46 (15.2) | 14 | 9.0 |
| 40–49 | 42 (13.9) | 11 | 9.0 |
| ≥50 | 62 (20.5) | 8 | 5.0 |
| **Sex** | | | |
| Male | 169 (56.0) | 11 | 9.0 |
| Female | 133 (44.0) | 10 | 8.0 |
| **Residence** | | | |
| Urban | 201(66.6) | 10 | 8.0 |
| Rural | 101 (33.4) | 11 | 10.0 |
| **Site of lesion(s)** | | | |
| Head and neck only | 270 (89.4) | 10 | 8.0 |
| Trunk and /or limbs only | 15 (5.0) | 8 | 6.0 |
| Multiple body regions | 17 (5.6) | 11 | 11.0 |
| **Clinical phenotype** | | | |
| LCL | 189 (62.6) | 9 | 8.0 |
| MCL | 104 (34.4) | 11 | 9.5 |
| DCL | 9 (3. 0) | 18 | 11.0 |

domain had the lowest percentage (16.6% and 25.0%, for females and males respectively). The HRQoL impairment was more pronounced in males compared to females in the 'leisure' (P = 0.002) and 'personal relationships' (p = 0.001) DLQI domains.

Individuals affected by DCL had significantly higher scores in the 'daily activities' (P = 0.018), 'personal relationship' (p = 0.002) and 'treatment' (p = 0.008) domains than those with other phenotypes (Table 5).

## Multivariate ordinal logistic regression

In the multivariate analysis, clinical phenotype of CL, age of participants and the affected body region remained significantly associated with DLQI scores (Table 6). The odds of having low HRQoL was eight times higher in DCL cases (P = 0.003) compared to those with LCL. Younger, 20 to 49 years, age groups and those having their head and face region affected had higher odds of having very poor HRQoL compared to those in 50 years and above, and those that have lesions on their Trunk and/or limbs. There was no significant difference in DLQI scores between male and female participants (P = 0.260) or rural and urban dwellers (P = 0.354).

**Table 2. Male and female participants with clinical phenotype of cutaneous leishmaniasis.**

| Sex of participants | Clinical phenotype (n, %) | | |
|---|---|---|---|
| | LCL | MCL | DCL |
| Male | 104 (55.03) | 58 (55.77) | 7 (77.78) |
| Female | 85 (44.97) | 46 (44.23) | 2 (22.22) |
| Total | 189 (100) | 104 (100) | 9 (100) |

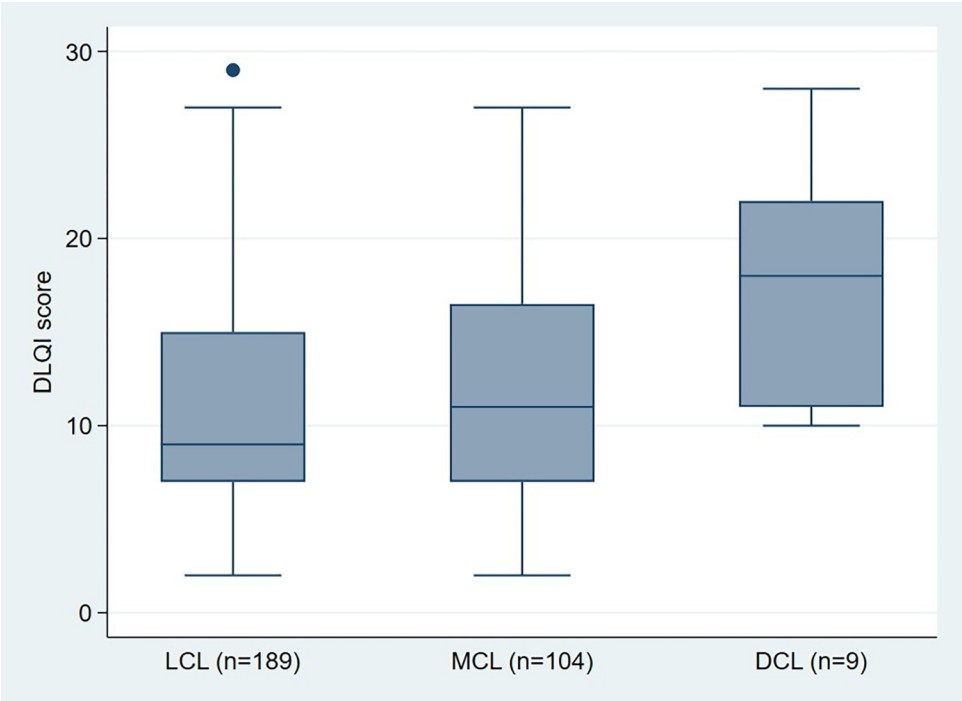

**Fig 1. DLQI score associated with clinical phenotype of cutaneous leishmaniasis.**

## Discussion

Skin diseases exert a considerable impact on social relationships, mental wellbeing and the activities. CL is a public health challenge in Ethiopia. The disease may cause severe disfigurement, because of destruction of anatomical structures and the formation of lifelong scars [8].

The concept of HRQoL encompasses physical activity, psychological well-being, the individual's degree of independence and their social relationships. Various studies showed the impact of CL in HRQoL of affected individuals [14,15,19].

Our study is the first of its kind assessing HRQoL in individuals with CL in Ethiopia. The median DLQI score was 10 (IQR 8). All participants with active CL experienced an effect on their HRQoL, which ranged from small effect to extremely large effect. Almost half of the participants reported either very large (36.4%) or extremely large (11.3%) effect on HRQoL. The size of effect associated with CL on HRQoL appears greater when compared to studies conducted in other parts of the world. A pilot study with 20 participants in Belo Horizonte, Brazil showed that 70% had a moderate/ large impact on their HRQoL [15]. A study in the United Kingdom of returning travelers indicated that 63% reported moderate or very large effect on their quality of life [16]. The high proportion of our participants (270, 89.4%) having a lesion

**Table 3. The size of effect on HRQoL associated with clinical phenotype of cutaneous leishmaniasis.**

| HRQoL effect | Clinical phenotype n (%) | | | Total N (%) |
| --- | --- | --- | --- | --- |
| | LCL | MCL | DCL | |
| Small | 25 (13.2) | 14 (13.5) | 0 (0) | 39 (12.9) |
| Moderate | 85 (45.0) | 33 (31.7) | 1 (11.1) | 119 (39.4) |
| Very large | 65 (34.4) | 41 (39.4) | 4 (44.4) | 110 (36.4) |
| Extremely large | 14 (7.4) | 16 (15.4) | 4 (44.4) | 34 (11.3) |

**Table 4. Percentage of total possible domain score and median/ Interquartile range (IQR) of DLQI domains scores by sex of participants.**

| DLQI domain | Percentage of possible score (Median, IQR) | | Rank Test | P value |
|---|---|---|---|---|
| | Female (n = 133) | Male (n = 169) | | |
| Symptoms and feelings (Q1,2) | 50.0 (3; 2–4) | 56.6 (3; 2–5) | Z 1.76 | 0.080 |
| Daily activities (Q3,4) | 33.3 (2; 0–3) | 35.0 (2; 0–3) | Z = 0.057 | 0.954 |
| Leisure (Q5,6) | 23.3 (1; 0–2) | 35.0 (2; 0–3) | Z = 3.079 | **0.002** |
| Work and school (Q7) | 73.3 (3; 1–3) | 66.6 (3; 1–3) | Z = -1.404 | 0.160 |
| Personal relationships (Q8,9) | 16.6 (0; 0–2) | 25.0 (1; 0–3) | Z = 3.191 | **0.001** |
| Treatment (Q10) | 30.0 (1; 0–1) | 33.3 (1; 0–2) | Z = 0.424 | 0.671 |

affecting the head and neck regions, might explain the large proportion of individuals with very large and extremely large HRQoL effects.

The individuals in the younger age group (20–49 years old) had greater odds of having low HRQoL compared to those aged 50 years or above age group which is a finding reported in individuals with psoriasis [20]. It may be that for younger individuals visible skin changes might exert a greater impact at a life stage when social roles or new relationships are developing.

Despite the differences in sociocultural grouping, study setting and causative *Leishmania* species, affected individuals are concerned with the extent of damage caused by CL. CL clinical phenotypes were associated significantly with different DLQI scores. Individuals with DCL had 8.8 times higher odds of having poor HRQoL compared to patients with LCL, which may be due to the extent of skin involvement and subsequent changes. Refai et al., observed that in Sri Lankan individuals with active LCL those with plaques and ulcerated lesions had higher DLQI scores than those with papules and nodules [21]. In Iran, Vares et al. found that those with ulcerated lesions had lower quality of life compared to those with non-ulcerated lesions [14].

We found that the "Leisure" (P = 0.002) and "Personal Relationship" (P = 0.001) DLQI domains significantly correlated with male sex. The reason for this is unclear and contrasts with a study of schizophrenia in which Ethiopian women had worse outcomes with respect to "life satisfaction" and "spousal relationships". [22]

The high scores in the "symptom and sign", and "work and school" DLQI domains associated with CL was seen in previous studies from Iran and Brazil [14,15]. The treatment domain appeared to be less affected which may be a result of our participants being assessed prior to treatment.

This study shows significant reduction in HRQoL for individuals with untreated CL in Ethiopia. A disease associated with this level of impairment requires effective, safe, acceptable and readily available treatment [23] which is not currently the case for many affected

**Table 5. Percentage of total DLQI domain score and median/ Interquartile range by clinical phenotype of participants.**

| DLQI domain | Percentage of possible score (Median, IQR) | | | Ch$^2$ | P value |
|---|---|---|---|---|---|
| | LCL (n = 189) | MCL (n = 104) | DCL (n = 9) | | |
| Symptoms and feelings (Q1,2) | 51.6 (3; 2–4) | 56.6 (3; 2–5) | 60.0 (4; 4–4) | 4.518 (df = 2) | 0.104 |
| Daily activities (Q3,4) | 32.3 (1; 0–3) | 37.5 (2; 1–3) | 60.0 (3; 3–5) | 7.972 (df = 2) | **0.018** |
| Leisure (Q5,6) | 28.3 (1; 0–3) | 31.6 (1; 0–3) | 50.0 (3; 0–6) | 2.604 (df = 2) | 0.272 |
| Work and school (Q7) | 70.0 (3; 1–3) | 70.0 (3; 1–3) | 50.0 (2; 1–3) | 0.911 (df = 2) | 0.634 |
| Personal relationships (Q8,9) | 18.3 (0; 0–2) | 26.6 (1; 0–3) | 43.3 (2; 2–4) | 11.711 (df = 2) | **0.002** |
| Treatment (Q10) | 30.0 (1; 0–2) | 26.6 (0.5; 0–2) | 73.3 (3; 2–3) | 9.586 (df = 2) | **0.008** |

**Table 6. Multivariate ordinal logistic regression of DLQI score with sociodemographic and disease characteristic of participants.**

| Variable | Median(IQR) | COR*(95%CI) | AOR**(95% CI) | P-Value (AOR) |
|---|---|---|---|---|
| **Age in years** | | | | |
| 18–19 | 9.5(10) | 1.9(1.0,3.7) | 1.8(0.9,3.5) | 0.082 |
| 20–29 | 11(8) | 2.2(1.3,3.8) | 2.1(1.2,3.8) | **0.007** |
| 30–39 | 14(9) | 3.7(1.9,7.4) | 3.6(1.8,7.3) | **<0.001** |
| 40–49 | 11(9) | 2.7(1.4,5.3) | 2.7(1.3,5.3) | **0.004** |
| 50 and above | 8(5) | 1 | 1 | |
| **Sex** | | | | |
| Male | 11(9) | 1.4(0.9,2.2) | 1.2(0.8,1.9) | 0.260 |
| Female | 10(8) | 1 | 1 | |
| **Residence** | | | | |
| Urban | 10(8) | 0.7(0.5,1.1) | 0.8(0.5,1.2) | 0.354 |
| Rural | 11(10) | 1 | 1 | |
| **Site of lesions** | | | | |
| Multiple body regions | 11(11) | 4.0(1.2,13.6) | 2.2(0.6,7.7) | 0.195 |
| Head and neck | 10(8) | 2.0(0.8,4.9) | 3.2(1.2,8.4) | **0.018** |
| Trunk and/or limbs | 8(6) | 1 | 1 | |
| **Clinical phenotype** | | | | |
| LCL | 9(8) | 1 | 1 | |
| MCL | 11(9) | 1.4(0.9,2.1) | 1.3(0.8,2.0) | 0.162 |
| DCL | 18(11) | 5.7(1.7,18.4) | 8.0(2.0,31.5) | **0.003** |

*COR = Crude Odds Ration

**AOR = Adjusted Odds Ration

individuals. Validated patient-reported outcome measures such as the DLQI are an important assessment of efficacy of treatments because of the long-term stigmatizing consequences of CL such as scarring.

## Limitations of the study

This study is a cross sectional prior to treatment, we cannot comment on possible treatment related changes in HRQoL. The clinical classification was based on the judgement of the dermatologist who assessed the patient rather than standardized case definitions.

## Acknowledgments

We are grateful to the study participants for their time and the dermatologists and staff of the Dermatology Department. Dr Michael Marks participated in discussions concerning analysis. A special thank you to Edom Getachew who assisted with data entry.

## Author Contributions

**Conceptualization:** Shimelis Doni, Kidist Yeneneh, Sophie Skarbek, Saba Lambert, Stephen L. Walker, Endalamaw Gadisa.

**Data curation:** Samuel Ayele, Abay Woday Tadesse, Endalamaw Gadisa.

**Formal analysis:** Yohannes Hailemichael, Mikyas Gebremichael, Samuel Ayele, Abay Woday Tadesse, Endalamaw Gadisa.

**Investigation:** Shimelis Doni, Kidist Yeneneh.

**Methodology:** Shimelis Doni, Kidist Yeneneh, Stephen L. Walker.

**Supervision:** Shimelis Doni.

**Writing – original draft:** Shimelis Doni, Kidist Yeneneh, Yohannes Hailemichael, Abay Woday Tadesse, Saba Lambert, Endalamaw Gadisa.

**Writing – review & editing:** Shimelis Doni, Kidist Yeneneh, Yohannes Hailemichael, Mikyas Gebremichael, Sophie Skarbek, Samuel Ayele, Saba Lambert, Stephen L. Walker, Endalamaw Gadisa.

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
