## [Decision Letter · Decision Letter 0]

11 Apr 2023

Dear Dr Gadisa,

Thank you very much for submitting your manuscript "Health-Related Quality of Life of Adults with Cutaneous Leishmaniasis at ALERT Hospital, Addis Ababa, Ethiopia" for consideration at PLOS Neglected Tropical Diseases. As with all papers reviewed by the journal, your manuscript was reviewed by members of the editorial board and by several independent reviewers. In light of the reviews (below this email), we would like to invite the resubmission of a significantly-revised version that takes into account the reviewers' comments. 

We cannot make any decision about publication until we have seen the revised manuscript and your response to the reviewers' comments. Your revised manuscript is also likely to be sent to reviewers for further evaluation.

Sincerely,

Walderez O. Dutra, PhD.

Section Editor

Walderez Dutra

Section Editor

Reviewer's Responses to Questions

**Key Review Criteria Required for Acceptance?**

**Methods**

-Are the objectives of the study clearly articulated with a clear testable hypothesis stated?

-Is the study design appropriate to address the stated objectives?

-Is the population clearly described and appropriate for the hypothesis being tested?

-Is the sample size sufficient to ensure adequate power to address the hypothesis being tested?

-Were correct statistical analysis used to support conclusions?

-Are there concerns about ethical or regulatory requirements being met?

Reviewer #1: This paper is clearly written, and the way the study is described should make it replicable. The population chosen is large enough to come to useful conclusions, and also to pinpoint differences based on gender (which turn out to be important in the conclusions).

Reviewer #2: 1) In the lines 104-106, the authors present all DLQI domain. However, in the results (Table 4) the treatment domain was not considered. This point must be detailed. Which moment the patients answered the questionnaire (before, during or after the treatment)? This is an important point that could affect the HRQoL of the patients, especially considering the CL-treatment limitations. This information could be better described in the methodology section.

Reviewer #3: The hypothesis and objectives are not clear. 

Was the inclusion consecutively? No information about it. 

Not all the variables are defined and explained example: sex and residence. 

How did they calculate the sample size?

Explain which groupings are chosen for comparisons and why example: between leishmaniasis types. 

How were missing data adressed? How much data was missing per variable?

Number of patients eligible, included, reasons for exclusion: Consider a flow diagram.

The multivariate model contains related variables (DCL and multiple body parts) and the arguments to enter variables are not mentioned in the methods. 

The last references are not well formatted.

**Results**

-Does the analysis presented match the analysis plan?

-Are the results clearly and completely presented?

-Are the figures (Tables, Images) of sufficient quality for clarity?

Reviewer #1: Results are clearly presented, the figures are helpful in visualising these differences.

Reviewer #2: 2) I suggest to the authors to show the median score of each DLQI domains according the clinical phenotype. Was any domain more affected by clinical forms?

Reviewer #3: Please explain what is COR and what is AOR in table 5

**Conclusions**

-Are the conclusions supported by the data presented?

-Are the limitations of analysis clearly described?

-Do the authors discuss how these data can be helpful to advance our understanding of the topic under study?

-Is public health relevance addressed?

Reviewer #1: The conclusions seem to flow very well from the findings, and are compared with literature in a useful way.

Reviewer #2: 3) I think that the conclusion presented should be more cautious since the HRQoL assessment was not carried out according to the treatment, or after the treatment.

Reviewer #3: The authors do not present a comparison with healthy, non CL, or cured patients. The authors have no background score for healthy individuals and therefore cannot conclude that the`The HRQoL impairment in people affected by CL is significant´

We have to remember that such scores are highly culturally dependent.

**Editorial and Data Presentation Modifications?**

Reviewer #1: There are some occasional errors in English usage, so the paper could benefit from a careful proofreading run (for example: see lines 199-202).

Reviewer #2: NA

Reviewer #3: Line 57 and 58 improved, counseling on the nature of CL, therapeutic options as well as clinical outcomes and complications. This sentence is not correct.

Table 1 Clinical fenotype DCL percentage contains an error.

**Summary and General Comments**

Reviewer #1: This is a useful study, and provides good evidence for use of the DLQI in assessing the life impact of CT and related conditions.

Reviewer #2: (No Response)

Reviewer #3: The manuscript is well written and presented and the topic very well chosen.

The lack of comparison with healthy patients impairs the interpretation of the results.

It would be essential to include such a control group in order to conclude how CL affects people.

The article might focus on the differences between groups such as: male/female, urban/rural, etc.

As it is now, the article doesnt provide novel information that helps to improve our understanding of CL related HRQL. 

Finally, qualitative interviews would enrich the interpretation of the findings.

PLOS authors have the option to publish the peer review history of their article (what does this mean?). If published, this will include your full peer review and any attached files.

Reviewer #1: No

Reviewer #2: No

Reviewer #3: No
---

## [Decision Letter · Decision Letter 1]

21 Jul 2023

Dear Dr Gadisa,

Thank you very much for submitting your manuscript "Health-Related Quality of Life of Adults with Cutaneous Leishmaniasis at ALERT Hospital, Addis Ababa, Ethiopia" for consideration at PLOS Neglected Tropical Diseases. As with all papers reviewed by the journal, your manuscript was reviewed by members of the editorial board and by several independent reviewers. In light of the reviews (below this email), we would like to invite the resubmission of a significantly-revised version that takes into account the reviewers' comments. 

Reviewer #3 feels that you have not adequately answered his concerns regarding your paper. Can you please address his comments specifically, taking into account the items on the STROBE checklist.

https://www.strobe-statement.org/checklists/

We cannot make any decision about publication until we have seen the revised manuscript and your response to the reviewers' comments. Your revised manuscript is also likely to be sent to reviewers for further evaluation.

Sincerely,

Charles L. Jaffe, Ph.D.

Section Editor

Walderez Dutra

Section Editor

Reviewer #3 feels that you have not adequately answered his concerns regarding your paper. Can you please address his comments specifically, taking into account the items on the STROBE checklist.

https://www.strobe-statement.org/checklists/

Reviewer's Responses to Questions

**Key Review Criteria Required for Acceptance?**

**Methods**

-Are the objectives of the study clearly articulated with a clear testable hypothesis stated?

-Is the study design appropriate to address the stated objectives?

-Is the population clearly described and appropriate for the hypothesis being tested?

-Is the sample size sufficient to ensure adequate power to address the hypothesis being tested?

-Were correct statistical analysis used to support conclusions?

-Are there concerns about ethical or regulatory requirements being met?

Reviewer #2: (No Response)

Reviewer #3: Comment: the hypothesis and objectives are not clear.

Response: The objective is stated on Lines 90-91. We have edited the sentence to read “We wished to assess

the HRQoL associated with CL in a referral hospital setting in Ethiopia using the DLQI.”

This comment was not responded. I miss a hypothesis. 

Comment: Was the inclusion consecutively? No information about it.

This comment has not been responded in the text. 

Comment: Not all the variables are defined and explained example: sex and residence.

Again this comment has not been responded. 

Comment: How did they calculate the sample size?

Again this comment is not responded in the main text. 

Comment: Explain which groupings are chosen for comparisons and why example: between leishmaniasis types.

They have not explained why they chose groups. 

Comment: How were missing data addressed?

They are not adressing missed data in the manuscript

Comment: How much data was missing per variable?

Again they are not adressing missed data in the manuscript

Comment: Number of Patients eligible, included, reasons for exclusion: Consider a flow diagram.

Such a flow diagram and description of patients eligible augments quality. That is lacking now. 

Comment: The multivariate model contains related variables (DCL and multiple body parts) and the arguments

to enter variables are not mentioned in the methods.

Again, this comment has not been responded. 

Comment: The last references are not well formatted.

This comment is responded.

**Results**

-Does the analysis presented match the analysis plan?

-Are the results clearly and completely presented?

-Are the figures (Tables, Images) of sufficient quality for clarity?

Reviewer #2: (No Response)

Reviewer #3: : Please explain what is COR and what is AOR in table 5: this comment has been responded.

**Conclusions**

-Are the conclusions supported by the data presented?

-Are the limitations of analysis clearly described?

-Do the authors discuss how these data can be helpful to advance our understanding of the topic under study?

-Is public health relevance addressed?

Reviewer #2: (No Response)

Reviewer #3: Comment: The authors do not present a comparison with healthy, non-CL, or cured patients. The authors have

no background score for healthy individuals and therefore cannot conclude that the `The HRQoL

impairment in people affected by CL is significant´ we have to remember that such scores are highly

culturally dependent.

I consider that a comparison of HRQoL results obtained under different cultural circunstances is not comparable. This is my main objection against the article.

**Editorial and Data Presentation Modifications?**

Reviewer #2: Dear Editor,

I would like to express my appreciation for submitting the article entitled "Health-Related Quality of Life of Adults with Cutaneous Leishmaniasis at ALERT Hospital, Addis Ababa, Ethiopia" and for allowing me to review it. After careful analysis of the manuscript and considering the responses provided by the authors, I am convinced that the work is well-structured, presents a significant study, and should be published in PLOS Neglected Tropical Diseases.

All of my comments were adequately addressed by the authors, and they made the necessary adjustments to the manuscript based on my suggestions. However, I would like to highlight an aspect that I believe could be further explored and elaborated upon in the discussion section: the choice of the tool used to assess the patients' quality of life.

Assessing the quality of life is a crucial component in studies aiming to understand the impact of diseases on patients' lives. I am pleased to see that the authors selected a specific tool for this purpose in their study. However, I believe it would be valuable if the authors dedicated more discussion space to describing the selection of the tool and discussing its advantages. Additionally, they could explore how this tool compares to other available options.

Once again, I would like to commend the authors for their work and thank you for the opportunity to review this article. I hope that my suggestions will be helpful in further improving the manuscript.

Sincerely,

Reviewer #3: (No Response)

**Summary and General Comments**

Reviewer #2: I think it would be important to discuss the tool used to assess quality of life and its limitations. Is the EQ-5D-3L more appropriate tool for this type of study?

Reviewer #3: My concerns have not been adressed.

PLOS authors have the option to publish the peer review history of their article (what does this mean?). If published, this will include your full peer review and any attached files.

Reviewer #2: No

Reviewer #3: Yes: Jacob Bezemer
---

## [Editor Report · Decision Letter 2]

21 Sep 2023

Dear Dr Gadisa,

We are pleased to inform you that your manuscript 'Health-Related Quality of Life of Adults with Cutaneous Leishmaniasis at ALERT Hospital, Addis Ababa, Ethiopia' has been provisionally accepted for publication in PLOS Neglected Tropical Diseases.

Best regards,

Charles L. Jaffe, Ph.D.

Section Editor

Walderez Dutra

Section Editor

Sorry it took so long to reach a final decision. Please note that I think there is a mistake on line 224, p. 13 of the discussion which should read, "symptoms and feelings," and not "symptom and sign."

---

## [Editor Report · Acceptance letter]

24 Oct 2023

Dear Dr Doni,

We are delighted to inform you that your manuscript, "Health-Related Quality of Life of Adults with Cutaneous Leishmaniasis at ALERT Hospital, Addis Ababa, Ethiopia," has been formally accepted for publication in PLOS Neglected Tropical Diseases.

Best regards,

Shaden Kamhawi

co-Editor-in-Chief

Paul Brindley

co-Editor-in-Chief
